# Two-Time Multiplexed Targeted Next-Generation Sequencing Might Help the Implementation of Germline Screening Tools for Myelodysplastic Syndromes/Hematologic Neoplasms

**DOI:** 10.3390/biomedicines11123222

**Published:** 2023-12-05

**Authors:** Oriol Calvete, Julia Mestre, Ruth M. Risueño, Ana Manzanares, Pamela Acha, Blanca Xicoy, Francesc Solé

**Affiliations:** 1MDS Group, Josep Carreras Leukaemia Research Institute, ICO-Hospital Germans Trias i Pujol, Universitat Autònoma de Barcelona, 08916 Badalona, Spain; 2Facultat de Biociències, Universitat Autònoma de Barcelona, 08193 Barcelona, Spain; 3Leukos Biotech, 08021 Barcelona, Spain; 4Faculty of Education, University of Atlántico Medio, 35017 Las Palmas, Spain; 5Hematology Service, ICO-Hospital Germans Trias i Pujol, 08916 Badalona, Spain

**Keywords:** myelodysplastic syndromes (MDS), genetic predisposition, targeted next-generation sequencing (tNGS), germline variants, genomic screening, multiplexed samples

## Abstract

Next-generation sequencing (NGS) tools have importantly helped the classification of myelodysplastic syndromes (MDS), guiding the management of patients. However, new concerns are under debate regarding their implementation in routine clinical practice for the identification of germline predisposition. Cost-effective targeted NGS tools would improve the current standardized studies and genetic counseling. Here, we present our experience in a preliminary study detecting variants using a two-time multiplexed library strategy. Samples from different MDS patients were first mixed before library preparation and later multiplexed for a sequencing run. Two different mixes including a pool of three (3×) and four (4×) samples were evaluated. The filtered variants found in the individually sequenced samples were compared with the variants found in the two-time multiplexed studies to determine the detection efficiency scores. The same candidate variants were found in the two-time multiplexed studies in comparison with the individual tNGS. The variant allele frequency (VAF) values of the candidate variants were also compared. No significant differences were found between the expected and observed VAF percentages in both the 3× (*p*-value 0.74) and 4× (*p*-value 0.34) multiplexed studies. Our preliminary results suggest that the two-time multiplexing strategy might have the potential to help reduce the cost of evaluating germline predisposition.

## 1. Introduction

The development of innovative and better cost-effective tools for the early detection of cancer is necessary to improve prevention, monitoring, and guidelines, as well as the design of less harmful therapeutic strategies to reduce the mortality rates of cancer patients [1]. The implementation of next-generation sequencing (NGS) techniques in diagnosis has uncovered the relevance of the molecular landscape leading the development of the disease. In particular, whole exome sequencing (WES) has contributed not only to gene discovery and the identification of new pathogenic pathways involved in cancer but also to the molecular characterization of patients, which has allowed the development of accurate classifications for risk stratification and personalized treatments. In this sense, more efficient targeted NGS (tNGS) tools for the evaluation of candidate genes are arising for better patient diagnoses and treatment guidance.

In myelodysplastic syndromes (MDS), while chromosomal aberrations are found in around 44% of patients [2], 90% of patients carry molecular alterations [3]. Thus, optimized cost-effective tools are needed for the management of MDS. To this end, tNGS tools are not only being implemented for uncovering molecular drivers but also for the detection of clonal hematopoiesis of indeterminate potential (CHIP) and the identification of germline predisposition to MDS. CHIP is defined as the presence of clonal mutations at low frequency in the peripheral blood (PB) of healthy individuals in genes recurrently mutated in myeloid neoplasms [4,5]. In this sense, recent studies have shown that cancer therapy shapes the fitness landscape of CHIP [6]. Therefore, the identification of CHIP in cancer patients has been suggested to be used as a predictive marker to identify patients at risk and prevent them from treatments that might trigger secondary MDS [7]. On the other hand, during the last few years, several genes mutated at the germline level have also been identified as predisposing factors to MDS [8]. For instance, germline mutations in *TP53*, *SAMD9*, *SAMD9L*, *SRP72*, and *TERT* [9], along with other myeloid genes such as *DDX41*, *GATA2*, *ANKRD26*, *ETV6*, *CEBPA*, *ASXL1*, and *RUNX1*, have been associated with the development of MDS [10], which are also described in certain inherited bone marrow (BM) failure syndromes [11]. Therefore, tNGS in the context of MDS is typically used for the identification of pathogenic clones at different VAFs with different clinical implications. To this end, the development of more sensitive and effective tNGS tools to identify patients at risk has not only improved diagnosis and predictive tools but has also improved the genetic counseling service and facilitated the identification of healthy donors for allogenic transplants. Currently, clinical laboratories and healthcare services have implemented NGS analysis in routine clinical practice to personalize the diagnosis, prognosis, treatment, and management of patients with MDS.

However, new challenges are arising regarding the clinical validation of these technologies, such as the interpretation of NGS data at the translational level. In this context, current efforts are being focused on evaluating the role of variants of unknown significance (VUS). Several guidelines have been published to facilitate the interpretation of obtained results [12,13,14]. The variant allele frequency (VAF) of CHIP related variants has been estimated to be between 2 and 10% in PB samples [15]. Otherwise, somatic and germline mutations causing detectable cytopenia are typically found to be between 10% and 30% VAF and greater than 30%, respectively. However, although a recent consensus has been achieved within the MDS community, new concerns regarding the age of onset in mutation carriers or the VAF thresholds are already under debate [16,17,18].

Moreover, although tNGS tools have reduced the cost of large-scale sequencing, several approaches for the adequate monitoring of patients are not fully included in the routine clinical practice of most healthcare centers of the world. The challenge of improving the detection techniques will guarantee a better evaluation of patients over time in the prevention and early detection of relapse and disease progression. To this end, DNA libraries from different individual samples are labeled and typically multiplexed in a unique sequencing run to optimize the cost per patient. Here, we present our experience detecting variants when samples are two-time multiplexed. For this study, different DNA samples were first pooled before library preparation, and the later different libraries were then multiplexed in a sequencing run (Figure 1). Thus, our study evaluates the percentage of detected variants in a two-time multiplexed library in comparison with the results derived from the same samples but individually sequenced (one-time multiplexed library). We consider that our experience might guide new studies to evaluate the utility of adding more samples per tNGS run and reduce the total cost per sample.

## 2. Materials and Methods

CD3+ lymphocytes were used as a reliable source of germline DNA in MDS patients. CD3+ lymphocytes from the PB of three MDS patients were isolated via immunomagnetic selection using autoMACS technology (Miltenyi Biotec, Cologne, Germany), followed by DNA extraction using the Maxwell RSC Cultured Cells DNA Kit (Promega, Madison, WI, USA).

CD3+ lymphocyte DNA samples from three MDS patients were individually sequenced via tNGS using a custom panel covering selected exons of 50 myeloid-related genes [19]. The custom panel was designed with the technical specifications of KAPA HyperCap Workflow 3.3 (Roche, Basilea, Switzerland) protocol based on the use of KAPA HyperCap Target Enrichment Probes, which allow for the enrichment of targeted regions through the hybridization capture strategy. Libraries were prepared by using 100 ng of genomic DNA from CD3+ lymphocyte samples using the KAPA HyperPlus Kit (Roche, Basilea, Switzerland). Unique dual-Indexed (UDI) primers were used for sample labeling before library multiplexing (Figure 1A). Sequencing was carried out using an Illumina MiSeq instrument with a 2 × 75 bp paired-end reads protocol for an average coverage of 1000×. The reads were aligned against the human reference genome (hg19/GRCh37) using BWA software (version 0.7.15-r1140). The SAMtools-1.10 and VarScan-2.4.0 program packages were used for variant calling. Annotation was performed using the ANNOVAR software (version 20200607).

Only variants with ≥100× region coverage and ≥25 reads for the alternative allele were considered for further analysis. The variants were filtered according to variant location (only variants found in exons or splicing sites were selected). Non-synonymous or frame-affecting pathogenic/likely pathogenic variants and minor allele frequency (MAF) values in the general population lower than 0.03 were considered for the selection of candidate variants.

The same three samples were sequenced again but including an additional pooling step before library construction (Figure 1B). The DNA concentration of each sample was quantified using a fluorometric method (Qubit, Thermo Fisher Scientific, Waltham, MA, USA). Equal mass amounts of DNA (250 ng) from each patient were mixed into a new tube (3× multiplex study) to obtain a final mix at a concentration of 50 ng/µL, thus ensuring that each patient’s DNA was equally represented in the pool. Finally, the DNA pool was adjusted in a new tube by mixing the corresponding DNA volume with 10 mM Tris–HCl to obtain a final mass of 100 ng, which was used for library preparation. Mixed DNA was labeled with the same UDI primer and included in a library multiplex. A second DNA pool including the same final amount of DNA (100 ng) but from four patients (the three previously sequenced patients and DNA from a non-previously sequenced patient for a 4× multiplex study) was prepared. Sequencing was carried out as previously described for the individual libraries.

Statistical analyses between the observed and expected VAFs were carried out using Fisher’s F test to compare the VAF variances and Student’s t-test to compare the average VAF between both groups.

## 3. Results

The purity rate achieved for the selection of CD3+ lymphocytes was, on average, 95% for the three samples. DNA from the CD3+ lymphocytes of the three MDS patients were sequenced via tNGS, and 232, 229 and 187 total variants were found in the variant calling. After filtering, 41, 30, and 30 variants annotated within exons were found in the studied patients, respectively. In total, 75 different variants were found, including 18 variants that were found in two or more patients and 57 that were only found in one patient (unique variants). Only six variants were considered as candidate variants in the three patients (Table 1).

The same DNA from the CD3+ lymphocyte samples of the selected patients was also sequenced via tNGS in the 3× and 4× multiplex studies. A total of 283 and 257 variants were found in the 3× and 4× multiplex studies, respectively. After filtering, 50 and 41 variants annotated in exons were found in the 3× and 4× multiplex studies, respectively. Only 40 out of the 75 total variants (53.3%) and 30 out of the 75 total variants (40%) found in the individual tNGS were also found in the 3× and 4× multiplexed studies, respectively (Table 1). However, the VAFs of the variants found in the individual tNGS not detected in the 3× and 4× multiplex studies were lower than 2.1 and 4.9, respectively, and/or with read depth below 25 in the individual tNGS; thus, these results were considered artifacts. Otherwise, 80% and 73.1% of the filtered variants found in the 3× and 4× multiplex studies were also found in the individual tNGS studies. The VAFs of the 10 variants found in the 3× and 4× multiplex studies that were not found in the individual tNGS were lower than 1.3% or had an altered sequencing depth; therefore, they were discarded. Finally, 75% of the variants detected in the 3× multiplex study were still detected in the 4× multiplex study.

The percentage of the filtered variants found in the individual tNGS and in the 3× and 4× multiplex studies was calculated per VAF range (Table 2). Only 46.48% and 28.17% of the filtered variants with VAF < 10% were found in the 3× and 4× multiplex studies, respectively. The 100% and the 96.55% of filtered variants with a VAF range between 10 and 30% and VAF > 30% ranges, respectively, were found in the 3× and 4× multiplex studies.

The VAF deviation between expected VAF and observed VAF per variant was also calculated in the multiplex studies (Table 3). Low-VAF variants might correspond with somatic alterations in the lymphoid lineage or cross contamination with the myeloid lineage, which was in agreement with the purity rates achieved for the CD3+ lymphocyte selection (95%). In any case, these variants were not included for further germline evaluation. The selected germline variants are specifically highlighted in Table 3. On average, observed VAF was only found to deviate from −1.75 to +1.47 and from −2.12 to +1.54 in the 3× and 4× studies, respectively. No significant differences were found between the observed and expected VAFs for the variants detected both in the individual tNGS and the 3× multiplexed (*p*-value of 0.98) and 4× multiplexed (*p*-value of 0.79) studies, which means that the observed VAFs in the multiplex studies are in agreement with the expected VAF estimations.

On the other hand, 100% of the six candidate variants found in the individual tNGS studies were also found in the two-time multiplexed studies (highlighted in grey in Table 3). The average VAF of the six candidate variants was 39.27%; thus, after pooling, the expected VAFs of the candidate variants were 13.25% and 9.94% in the 3× and 4× multiplex studies, respectively. The observed VAFs of the candidate variants were 12.42% and 8.27% in the 3× and 4× multiplex studies, meaning there was an average VAF detection deviation of −0.83% and −1.67%, respectively. The VAF deviation ranged from −1.37% to 1.83% in the 3× multiplex study and from −2.12% to 0.2% in the 4× multiplex study (Table 3). No significant difference in VAF was found between the observed and expected VAFs of the candidate variants both in the 3× (*p*-value of 0.74) and 4× multiplex (*p*-value of 0.34) studies.

## 4. Discussion

Our pilot study shows that 100% of the candidate variants were detected in both the 3× and 4× multiplexing studies. Thus, up to four samples from different patients may be pooled before multiplexing without affecting the efficacy of candidate variants detection. In this sense, a non-significant deviation was found between the observed and expected VAFs of candidate variants both in the 3× and 4× multiplex studies. Nevertheless, the observed VAF for the candidate variants might fall down below 10% in two-time multiplex studies, which is especially relevant for somatic variant detection. For example, the candidate variant in the NRAS gene with a 19.5% VAF found in patient 2, was detected at VAFs of 4% and 3.4% in the 3× and 4× multiplex studies, respectively. Therefore, low VAF variants should not be ruled out even when searching for germline predisposition. Thus, our preliminary results suggest that the two-time multiplexing strategy might be a useful tool for germline predisposition evaluation (high-VAF variants). Taking this into account, it is important to highlight that germline mutations could explain at least 15% of adult and pediatric MDS [20]. In specific contexts, such as adolescents with MDS and monosomy of chromosome 7, this percentage could reach up to 70% [21]. In addition, molecular alterations involve not only germline predisposition to MDS but other scenarios such as therapy-related myeloid neoplasms [22] or MDS in elderly patients [16]. Furthermore, several hereditary syndromes have also been associated with the development of MDS [23], where the MDS arise within the genetic landscape context that predisposes to multiple tumors including BM failure [24]. The accurate identification of patients with germline predisposition is mandatory since their clinical management differs from sporadic MDS [12]. Thus, reducing the cost of tNGS tools in patient diagnosis will improve the routine clinical practice of healthcare services.

On the other hand, lower efficacy for variant detection was observed for the identification of filtered variants with VAF < 10% when multiplexing. Nevertheless, most of the variants found in this VAF range were considered sequencing artifacts. However, a significant decrease in efficacy in variant detection must be assumed in low VAF detection studies, such as CHIP identification, where the candidate variants are found in a VAF range between 2 and 10%.

In summary, two-time multiplexing must serve to detect molecular alterations at a reduced cost. It is important to note that a Sanger sequencing validation must be performed after multiplexing in the individual samples mixed in the pool to identify the variant carrier. However, primer designing and the Sanger sequencing protocol are a more than 10 times cheaper to carry out compared to individual tNGS. In addition, a reduced number of validations are expected when searching for germline predisposition. Moreover, only 5% to 15% of adults and 4% to 13% of pediatric patients with MDS or AML, respectively, carry germline pathogenic variants in cancer susceptibility genes [25]. Likely, the number of different variants and genes involved in the development of MDS is limited, which means that same Sanger primers can be reused in different sequencing runs. Finally, a bioinformatic analysis must be performed for each sample when sequencing. However, only one bioinformatic analysis instead of three is needed when pooling in 3× multiplexed studies, reducing the costs and time needed to carry out the protocol.

In any case, the proposed cost-effective tNGS strategy is recommended for the large-scale screening of genetic predisposition. The development of cost-effective tNGS tools will not only improve the current standardized studies but also lead to improvements when searching for germline MDS predisposition diagnosis in candidate patients. In addition, although germline predisposition is typically described in adolescents and young adults in the upper age range of 40 years, it is important to highlight that a genetic predisposition to MDS can also be present in older adulthood [23]. Germline mutations in the *DDX41* gene, telomere biology disorders, *GATA2*, and thrombocytopenia-associated disorders are associated with myeloid malignancies in older adults [26]. Thus, the criteria of older age at the onset of MDS is insufficient to rule out germline predisposition, which is especially important when searching for non-familiar donors that must be tested for pathogenic variants. Reducing costs of tNGS studies will allow for the extensive screening of individuals when searching for healthy donors for hematopoietic stem cell transplantation purposes [26].

Our preliminary study could inspire new trials to achieve higher standards. Future studies must be performed to improve the low VAF variant detection in two-time multiplexing tNGS. Importantly, additional studies including different sequencing platforms, library preparation protocols, and labeling techniques must be performed. In addition, in the present study, germline DNA was obtained from CD3+ lymphocytes. However, when using DNA from CD3+ cells, early somatic changes occurring in immature hematopoietic stem cells (HSCs) might drive false positive germline predisposition detection. Alternatively, hair follicles or cultured fibroblasts would be a preferable source of germline DNA in patients with MDS when possible. Finally, two-time multiplexing strategies must be exported to other tumor types to widen the advantages of multiplexing in metastatic diseases and liquid biopsy strategies. In these studies, total PB might be used instead of DNA from CD3+ lymphocytes, simplifying germline predisposition screening.

## Figures and Tables

**Figure 1 biomedicines-11-03222-f001:**
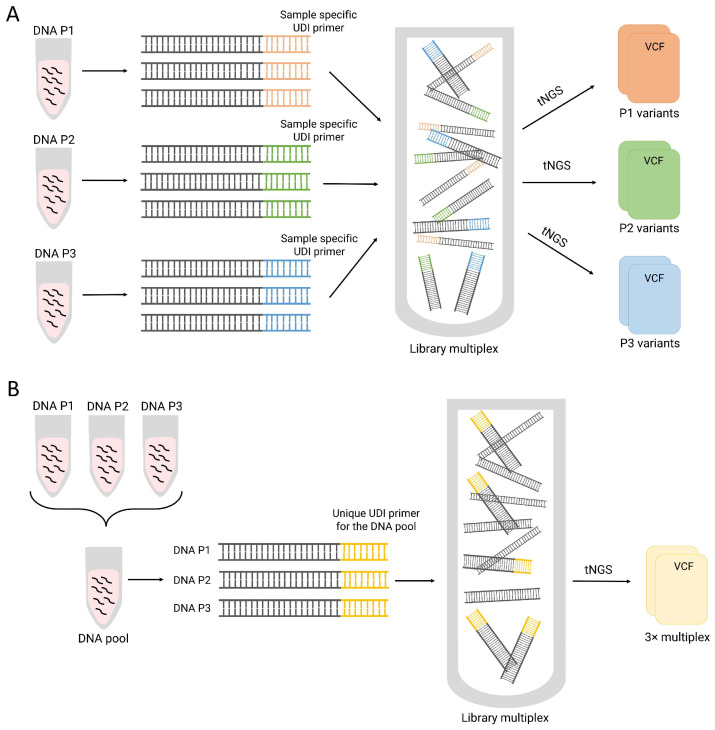
Sequencing strategies. (**A**) CD3+ lymphocyte DNA samples from three MDS patients were individually sequenced via tNGS. Three different UDI primers were used to label each sample. Variants were filtered from a VCF file per sample (**B**) CD3+ lymphocyte DNA samples were pooled before library preparation and multiplexing. The same UDI primer was used to label the pooled DNA from the three patients. Variants were filtered from a unique VCF file for all samples. P: Patient. VCF: Variant Call Format. UDI: Unique Dual Indexed. tNGS: targeted next generation sequencing.

**Table 1 biomedicines-11-03222-t001:** Number of variants detected per patient in individual tNGS and multiplex studies.

		Filtered Variants					Candidate Variants		
Patient	Variants	tNGS	3× M	(%)	4× M	(%)	tNGS	3× M	4× M
1	232	41	24	58.54	20	48.78	1	1	1
2	229	30	20	66.67	14	46.67	0	0	0
3	187	30	18	60.00	13	43.33	5	5	5
Total (different variants)	350	75	40	53.33	30	40.00	6	6	6

tNGS: targeted next generation sequencing, M: Multiplex; (%): Percentage of variants from the tNGS studies found in the multiplex study.

**Table 2 biomedicines-11-03222-t002:** Percentage of filtered variant detection per VAF range.

VAF Range	Patient	tNGS	3× M	(%)	4× M	(%)
(<10%)	1	30	13	43.33	13	43.33
	2	24	14	58.33	13	54.17
	3	17	6	35.29	4	23.53
	TOTAL	71	33	46.48	20	28.17
(10–30%)	1	0	0	NA	0	NA
	2	0	0	NA	0	NA
	3	1	1	100.00	1	100.00
	TOTAL	1	1	100.00	1	100.00
(>30%)	1	11	11	100.00	11	100.00
	2	6	6	100.00	6	100.00
	3	12	11	91.67	11	91.67
	TOTAL	29	28	96.55	28	96.55

tNGS: targeted next generation sequencing, M: Multiplex; (%): Percentage of variants from the tNGS studies found in the multiplex study.

**Table 3 biomedicines-11-03222-t003:** Filtered variants per patient. Candidate variants are highlighted in grey.

	Patient			3× Multiplex			4× Multiplex		
Variant	1	2	3	Expected VAF	Observed VAF	VAF Difference	Expected VAF	Observed VAF	VAF Difference
TET2:p.C1135W	3.8	0	0	1.27	1.90	0.63	0.95	ND	ND
TET2:p.L1819 *	4.9	0	0	1.63	2.10	0.47	1.23	ND	ND
TET2:p.P401del	0	0.7	0	0.24	0.50	0.26	0.18	ND	ND
TET2:p.G613W	0	1.4	0	0.47	1.10	0.63	0.35	ND	ND
TET2:p.G614E	0	1.3	0	0.43	0.94	0.50	0.33	ND	ND
TET2:p.L615P	0	1.3	0	0.43	0.93	0.49	0.33	ND	ND
TET2:p.P616P	0	1.3	0	0.43	0.95	0.51	0.33	ND	ND
DDX41:p.R400R	46.4	47.8	0	31.40	49.10	17.70	23.55	60.50	36.95 †
CUX1:p.P1439P	55.2	0	49.4	34.87	35.30	0.43	26.15	55.00	28.85 †
CUX1:p.S1448del	3.8	1.9	2.3	2.67	3.90	1.23	2.00	3.50	1.50
CUX1:p.R558Q	2.6	1.5	0	1.37	0.96	−0.40	1.03	2.10	1.08
EZH2:p.E401Kfs*22	5.1	0	0	1.70	1.40	−0.30	1.28	1.40	0.13
EZH2:p.P132P	53.6	0	0	17.87	16.80	−1.07	13.40	11.80	−1.60
EZH2:p.M1?	5.1	0	0	1.70	1.80	0.10	1.28	ND	ND
EZH2:p.D185H	0	0	99.9	33.30	25.90	−7.40	24.98	18.40	−6.58
ANKRD26:p.Q20R	51.6	50.7	49.7	50.67	48.10	−2.57	38.00	50.20	12.20
CBL:p.D460del	3.7	3.9	3	3.53	3.20	−0.33	2.65	3.40	0.75
TP53:p.S366A	51.2	0	0	17.07	18.90	1.83	12.80	13.00	0.2
NF1:p.L234L	99.9	99.8	49.3	83.00	86.30	3.30	62.25	89.50	27.25 †
CEBPA:p.T230T	53.5	0	0	17.83	18.80	0.97	13.38	14.20	0.82
CEBPA:p.P189del	2.9	4	4.9	3.93	4.60	0.67	2.95	5.30	2.35
CEBPA:p.G104del	1.9	1.9	0	1.27	1.50	0.23	0.95	1.70	0.75
ASXL1:p.Q428Sfs*10	6.3	0	0	2.10	1.50	−0.60	1.58	1.50	−0.08
ASXL1:p.G645Vfs*58	2.8	2.4	1.7	2.30	2.60	0.30	1.73	2.80	1.08
ASXL1:p.G646Wfs*12	1.2	1.7	35.4	12.77	11.80	−0.97	9.58	7.80	−1.775
ASXL1:p.T655T	49.1	0	0	16.37	16.00	−0.37	12.28	12.50	0.23
ASXL1:p.L1325F	51	0	0	17.00	17.20	0.20	12.75	13.10	0.35
BCOR:p.D420D	99.9	46.3	99.6	81.93	78.60	−3.33	61.45	67.30	5.85
BCORL1:p.E1324del	0.7	0.7	0	0.47	0.87	0.40	0.36	0.57	0.22
JAK2:p.V617F	0	7.5	0	2.50	1.70	−0.80	1.88	1.80	−0.08
IDH2:p.V109V	0	46.9	0	15.63	16.60	0.97	11.73	11.90	0.18
NRAS:p.Q61H	0	0	5.6	1.87	2.10	0.23	1.40	ND	ND
NRAS:p.G13R	0	0	19.5	6.50	4.00	−2.50	4.88	3.40	−1.48
NRAS:p.G12C	0	0	1.8	0.60	0.76	0.16	0.45	ND	ND
GATA2:p.P5P	0	46.9	0	15.63	17.90	2.27	11.73	14.50	2.78
GATA2:p.R396Gfs*81	0	0	44.3	14.77	14.60	−0.17	11.08	8.80	−2.28
SRSF2:p.P95H	0	0	41.8	13.93	13.10	−0.83	10.45	7.30	−3.15
SETBP1:p.D868N	0	0	43.4	14.47	12.10	−2.37	10.85	9.30	−1.55
SETBP1:p.H1206H	0	0	50.6	16.87	12.10	−4.77	12.65	10.00	−2.65
ZRSR2:p.P303P	0	0	46.7	15.57	14.60	−0.97	11.68	11.70	0.02
Average of total negative VAF deviation						−1.75			−2.12
Average of total positive VAF deviation						1.47			1.54
Average VAF of candidate variants		39.4		13.25	12.42	−0.83	9.94	8.27	−1.67
Average of candidate variants negative VAF deviation						−1.37			−2.05
Average of candidate variants positive VAF deviation						1.83			0.20

ND (Not detected): Variants not appearing in the 4× multiplex. ?: variant affecting the start codon. *: variant affecting the frame that include a premature stop codon. †: not included in the average VAF deviation because this variant was found in patient 4. Expected VAF: Average VAF obtained from the summatory of VAF in the individual tNGS divided into the number of samples added into the multiplex. Observed VAF: VAF observed in the multiplex file.

## Data Availability

The data presented in this study are available on request from the corresponding author.

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
