# Peer review of "Two-Time Multiplexed Targeted Next-Generation Sequencing Might Help the Implementation of Germline Screening Tools for Myelodysplastic Syndromes/Hematologic Neoplasms"

_biomedicines, 2023, doi:10.3390/biomedicines11123222_

Round 1

Reviewer 1 Report

Comments and Suggestions for Authors

I do not understand the approach.

The study is supposed to facilitate diagnosis of germline variants potentially associated with MDS. The material is supposed to represent germline. Why then, study,report and analyses multiple variants with low VAFs? That do not seem germline at all?

I also do not agree there is a need to do universal variant screening in candidate family donors for HSCT? Why not just test the variant(s) found (germline) in MDS patients? why do not really expect a relative of a person with germline variant related MDS would have another variant known to be related with MDS.

Also the paper does not mention the costs of bionformatic analysis which is higher if the system gets more complicated.

Comments on the Quality of English Language

No major problems

Reviewer 2 Report

Comments and Suggestions for Authors

In the current manuscript, the authors analyzed the utility of pooled targeted-sequencing for the detection of genetic alterations. The experiments are well performed and the manuscript is well written. Here are my comments that would improve the manuscript.

1.           It would be helpful to have a figure that summarizes the methods so that the readers can understand the current study.

2.           The authors state that they adjusted the DNA to a total of 100 ng from the three cases. However, I think they could not adjust it to 100 ng using equal amounts of DNA from the three cases.  It would be better to describe the method in more detail.

3.           What is the “MAF” in line 109 ?

4.          Although the title suggests that the authors analyzed germline predisposition, they actually seemed to identify some somatic mutations as in Table 3. These points are confusing for the readers.

Round 2

Reviewer 1 Report

Comments and Suggestions for Authors

The paper is satisfactory for publication now.

Comments on the Quality of English Language

No major problems.